# Feasibility study of a randomised controlled trial to investigate the treatment of sarcoidosis-associated fatigue with methylphenidate (FaST-MP): a study protocol

Christopher Atkins,[1,2] Richard Fordham,[1] Allan B Clark,[1] Andrea Stockl,[1] Andrew P Jones,[1] Andrew M Wilson[1,2]

[1]Department of Medicine, University of East Anglia Norwich Medical School, Norwich, Norfolk, UK
[2]Department of Respiratory Medicine, Norfolk and Norwich University Hospital, Norwich, Norfolk, UK

**Correspondence to**
Dr Christopher Atkins;
c.atkins@uea.ac.uk

## ABSTRACT

**Introduction** Fatigue is a frequent and troublesome manifestation of chronic sarcoidosis. This symptom can be debilitating and difficult to treat, with poor response to the treatment. Symptomatic management with neurostimulants, such as methylphenidate, is a possible treatment option. The use of such treatment strategies is not without precedent and has been trialled in cancer-related fatigue. Their use in sarcoidosis requires further evaluation before it can be recommended for clinical practice.

**Methods and analysis** The Fatigue and Sarcoidosis—Treatment with Methylphenidate study is a randomised, controlled, parallel-arm and feasibility trial of methylphenidate for the treatment of sarcoidosis-associated fatigue. Patients are eligible if they have a diagnosis of sarcoidosis, significant fatigue (measured using the Fatigue Assessment Scale) and have stable disease. Up to 30 participants will be randomly assigned to either methylphenidate (20 mg two times per day) or identical placebo in a 3:2 ratio for 24 weeks. The primary objective is to collect data determining the feasibility of a future study powered to determine the clinical efficacy of methylphenidate for sarcoidosis-associated fatigue. The trial is presently open and will continue until July 2018.

**Ethics and dissemination** Ethical approval for the study was granted by the Cambridge Central Research Ethics Committee on 21 June 2016 (reference 16/EE/0087) and was approved and sponsored by the Norfolk and Norwich University Hospital (reference 190280). Clinical Trial Authorisation (EudraCT number 2016-000342-60) from the Medicines and Healthcare products Regulatory Agency (MHRA) was granted on 19 April 2016. Results will be presented at relevant conferences and submitted to appropriate journals following trial closure and analysis.

**Trial registration number** NCT02643732; Pre-results.

## INTRODUCTION

Sarcoidosis is a granulomatous, multisystem disease with an incidence of 5 per 100 000 person–years, typically affecting patients of working age.[1] Fatigue has been described as a 'core symptom' of sarcoidosis and is present in up to 80% of patients.[2 3] The presence of fatigue has shown to adversely affect quality of life.[4] Sarcoidosis-associated fatigue has multiple possible aetiologies. These include subclinical disease activity,[5] increased circulating tumour necrosis factor-alpha levels,[6] disturbed circadian rhythm[7] and sleep,[8] small-fibre neuropathy,[9] depression,[10] sleep-disordered breathing problems,[11] treatment-related side effects[12] or a combination of any of these factors.

### Role of neurostimulants in treating fatigue

Current evidence for treatments of sarcoidosis-associated fatigue is weak, with further work required.[13] The use of methylphenidate for fatigue is not without precedent, having been trialled in cancer-related fatigue and recommended as effective for symptomatic relief after meta-analysis by the Cochrane collaboration.[14] There is also evidence for effect in fatigue associated with Parkinson's disease[15] and HIV.[16] In all

**Strengths and limitations of this study**

► Twenty-four-week duration of receiving methylphenidate or placebo.
► Variety of outcome measures included to inform future study design.
► Exercise measurements during study done both in clinic (modified shuttle walk test) and using objective measures of exercise during free-living (wrist-worn activity monitors).
► Small participant numbers will not prove clinical efficacy of methylphenidate but does allow appropriate future study design to be determined.
► No stratification for age or sex due to small number of participants.

cases, the treatment has only been trialled for a short period of time (up to 12 weeks). Neurostimulants have been trialled for symptomatic relief of fatigue associated with sarcoidosis. Both methylphenidate[17] and modafinil[18] have been trialled in patients with significant fatigue, though in only a very small sample (10 patients trialled methylphenidate and 15 trialled modafinil) over only 8 weeks, a period of time that does not reflect the typical period of time that these agents may be used for in clinical practice. Despite initial evidence for the use of methylphenidate in sarcoidosis-associated fatigue being published almost 10 years ago, no further studies have been published; the reasons for this are unclear when sarcoidosis-associated fatigue remains a significant clinical issue without established treatment strategies. Questions remain about ongoing treatment effect, side effects during protracted treatment and whether patients would continue to take the medication over a longer period. For these reasons, this feasibility study has been undertaken prior expanding to a large, multi-centre randomised controlled trial (RCT).

### Rationale for this pilot trial

High quality evidence is required to establish the clinical efficacy and long-term tolerability of methylphenidate for the treatment of sarcoidosis-associated fatigue. Change in fatigue scores measured by a validated questionnaire measure of fatigue across a longer treatment period of 6 months would be able to address the issue of efficacy, but the evidence to support a large-scale RCT, or determine the optimal design of such an RCT, is presently insufficient.

Furthermore, although recommendations from the World Association of Sarcoidosis and Other Granulomatous diseases recommend measuring fatigue in trials involving patients with sarcoidosis using a validated questionnaire,[19] we also plan to investigate the effect on activity levels with treatment using wrist-worn accelerometers alongside typical questionnaire measures. Outputs from these activity monitors appear to correlate with fatigue scores, both in terms of reduced activity and increased time in sedentary behaviours (C Atkins, unpublished data, 2016), and may give further insight into how improvements in fatigue affect day-to-day living in patients. To plan a sufficiently large RCT, we propose a pilot placebo-controlled RCT to inform the design of a future, definitive trial investigating the management of fatigue.

### OBJECTIVES

The primary hypothesis is to determine the feasibility and design of a future, large-scale RCT investigating the use of methylphenidate to treat sarcoidosis-associated fatigue and provide sustained benefit over a 6-month period. In addition, it will pilot the use of wrist-worn, triaxial accelerometer-based activity monitors to monitor change in activity and sedentary behaviours during the treatment alongside validated questionnaire-based fatigue scores.

### METHODS

#### Participants

This study is a randomised, double-blind, placebo-controlled single-centre parallel-group trial with randomisation of up to 30 patients in a ratio of 3:2 in favour of methylphenidate. This unequal arm size will increase data collected from the treatment arm, including adverse event rates. The trial is currently taking place at a single tertiary centre (Norfolk and Norwich University Hospital (NNUH), UK) and will remain open until July 2018. Participants are identified through the respiratory clinic at the NNUH. The intervention is methylphenidate, initially 10 mg two times per day and increased to 20 mg two times per day, if appropriate, compared with an identical placebo. Participants and outcome assessors are blinded to the treatment received. Allocation is through an on-line system which allocates participants to methylphenidate or placebo and automatically informs the hospital pharmacy of the group allocated to allow dispensing of treatment while maintaining blinding. Participant flow through the trial is shown in figure 1.

Study participants are required to attend eight clinical visits over a 26-week period; one screening visit, a review at baseline, assessments for side effects at 2 and 4 weeks, assessment of clinical outcomes at 6, 12, 18 and 24 weeks. There is one post-trial assessment 6 weeks after discontinuing. Between visits, participants are contacted by phone call to monitor for side effects. Study visits and activities take place at the NNUH, although patients treated by other hospitals in the East of England region can be recruited if their centre is willing to act as a Participant Identification Centre (PIC site) and participants are willing to travel to the NNUH. The overview of study visits, including the assessments performed, is shown in table 1.

All participants must have a diagnosis of sarcoidosis. The full list of inclusion and exclusion criteria are shown in box 1. All eligible participants with sarcoidosis and fatigue (defined as a Fatigue Assessment Scale (FAS) score more than 21) who are invited to participate in the study are provided with written information. Participants who agree to participate are required to give written informed consent. Initial contact with participants is during their attendance at the respiratory outpatient department or following referral from the patient's primary respiratory physician.

#### Interventions

A screening visit occurs 2 weeks prior to commencing study medications. This visit is used to collect written informed consent, baseline demographic data, list current medications (including current and prior treatment for sarcoidosis), measure blood pressure, pulse and perform a baseline ECG and blood tests (renal function

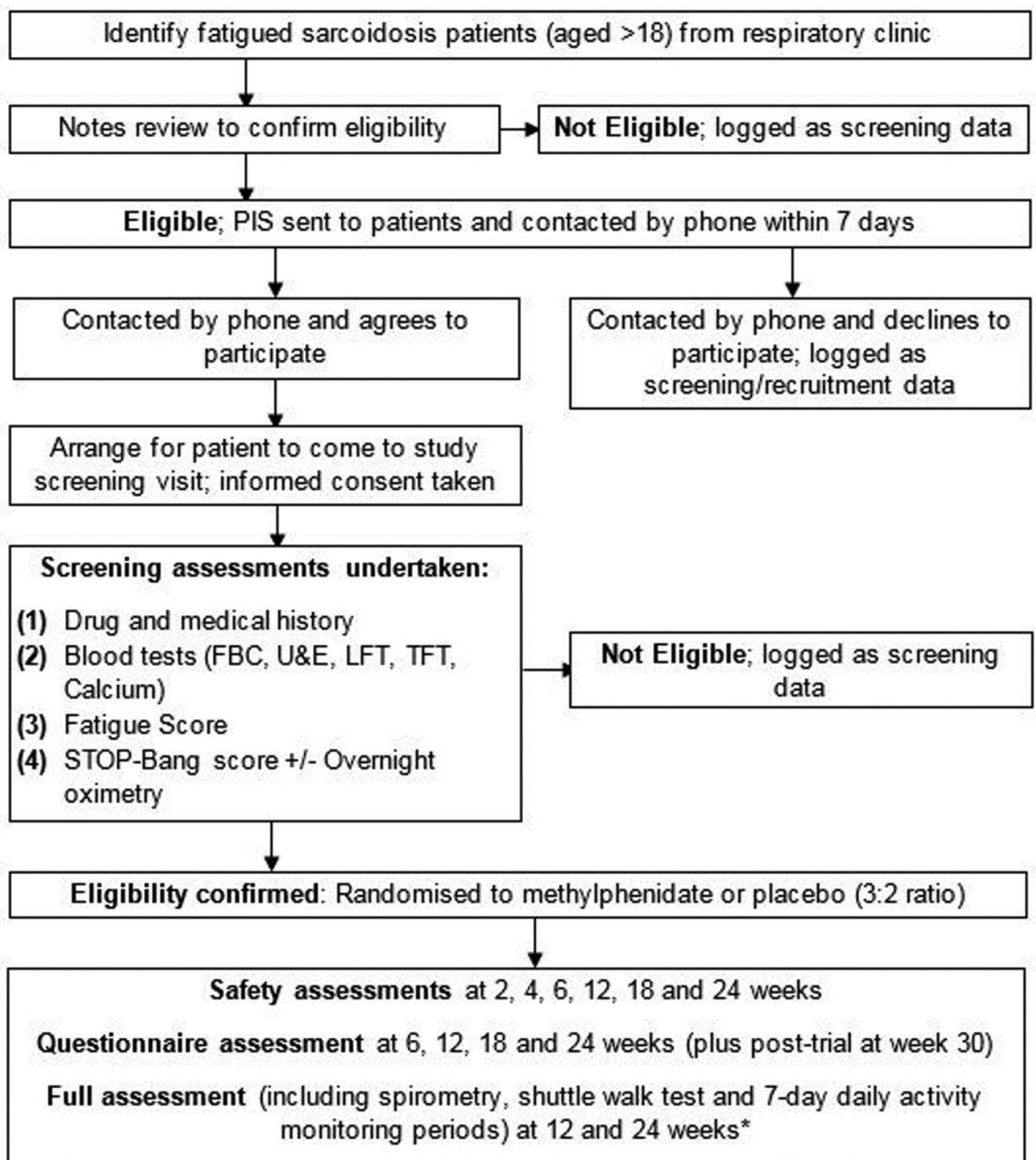

**Figure 1** Patient flow through trial. FBC, full blood count; LFT, liver function test; PIS, patient information sheet; TFT, thyroid function test; U&E, urea and electrolytes.

and electrolytes, liver function and full blood count). Female participants undertake a pregnancy test. Clinical data are reviewed for manifestations of sarcoidosis, to determine whether extra pulmonary disease is present. Fatigue is quantified by FAS questionnaire; all participants must have a baseline FAS score of more than 21 points to participate. Obstructive sleep apnoea (OSA) is screened for using the STOP-Bang Questionnaire. Participants scoring above 4 points undergo overnight pulse oximetry to determine if OSA is present, those with a desaturation index of >15 being excluded and referred for consideration of the treatment. Finally, potential participants wear an activity monitor (GENEActiv Original; Activinsights,

Cambridgeshire, UK) on their non-dominant wrist for a 7-day period to measure daily activity levels prior to commencing therapy. Once baseline fatigue, comorbidities, medications and blood tests have been checked to confirm no contraindication to participation, the subject is randomised.

When participants have been confirmed as eligible for participation in the Fatigue and Sarcoidosis–Treatment with Methylphenidate (FaST-MP) study following their screening visit, they are invited back 2 weeks later to undertake the baseline visit which records preintervention questionnaire scores and spirometry. At this visit, participants complete fatigue scores (FAS score and the Functional

**Table 1** Study overview (SPIRIT template)

| Timepoint | Enrolment | Allocation | Postallocation | | | | | | Close out |
|---|---|---|---|---|---|---|---|---|---|
| | Study period | | | | | | | | |
| | −2 weeks | 0 weeks | 2 weeks (±3 days) | 4 weeks (±3 days) | 6 weeks (±3 days) | 12 weeks (±1 week) | 18 weeks (±1 week) | 24 weeks (±1 week) | +4–8 weeks |
| **Enrolment:** | | | | | | | | | |
| Eligibility screen | X | | | | | | | | |
| Informed consent | X | | | | | | | | |
| Allocation | | X | | | | | | | |
| **Interventions:** | | | | | | | | | |
| *Methylphenidate* (X=uptitrate dose) (O=drug dispensed) | | O | O X | O | O | O | O | | |
| *Placebo* (X=uptitrate dose) (O=drug dispensed) | | O | O X | O | O | O | O | | |
| **Assessments** | | | | | | | | | |
| Safety bloods (FBC/LFT/U+Es) | X | | X | X | X | X | | | |
| Safety questionnaire | | X | X | X | X | X | X | X | |
| Pregnancy test | X | X | X | | | | | | |
| ECG | X | | X | X | X | X | | X | |
| Spirometry | | X | | | | X | | X | |
| MSWT | | X | | | | X | | X | |
| Accelerometer (7 days) | X | | | | | X | | X | |
| **Questionnaires** | | | | | | | | | |
| FAS | X | X | X | X | X | X | X | X | X |
| FACIT-F | | X | X | X | X | X | X | X | X |
| HADS | | X | X | | X | X | X | X | X |
| SF-36 | | X | X | | X | X | X | X | X |
| EQ5D | | X | X | | X | X | X | X | X |
| KSQ | | X | X | | X | X | X | X | X |
| Costs | | X | | | | X | | X | X |
| PSQI | | X | | | | X | | X | X |

Continued

**Table 1** Continued

| Timepoint | Study period | | Postallocation | | | | | | Close out |
|---|---|---|---|---|---|---|---|---|---|
| | Enrolment | Allocation | 2 weeks | 4 weeks | 6 weeks | 12 weeks | 18 weeks | 24 weeks | |
| | –2 weeks | 0 weeks | (±3 days) | (±3 days) | (±3 days) | (±1 week) | (±1 week) | (±1 week) | +4–8 weeks |
| Exit questionnaire | | | | | | | | X | |
| Focus group (post-trial) | | | | | | | | | X |

In addition, all participants receive telephone calls at weeks 1, 3, 5, 8, 10, 14, 16, 10 and 22 to review safety (emergence of side effects).
EQ5D, EuroQoL-5D-5L; FACIT-F, Functional Assessment of Chronic Illness Therapy – Fatigue; FAS, Fatigue Assessment Scale; FBC, full blood count; HADS, Hospital Anxiety and Depression Score; KSQ, King's Sarcoidosis Questionnaire; LFT, liver function test; MSWT, modified shuttle walk test; PSQI, Pittsburgh Sleep Quality Index; SF-36, Short Form-36; SPIRIT, Standard Protocol Items: Recommendations for Interventional Trials; U+Es; urea + electrolytes.

---

**Box 1   Eligibility criteria**

**Inclusion criteria**
► A proven diagnosis of sarcoidosis: this is defined as either a biopsy-proven disease (non-caseating granulomas from a tissue biopsy) or a diagnosis of sarcoidosis agreed by an interstitial lung disease multidisciplinary team meeting.
► Stable disease (treatment unchanged for 6 weeks, without anticipation of change in treatment during trial period).
► Able to give informed consent.
► Fatigue Assessment Scale score greater than 21 units.

**Exclusion criteria**
► Evidence of coexisting obstructive sleep apnoea.
► Documented history of significant cardiac disease (including cardiac sarcoid) or associated disease which would increase risk of underlying coronary artery disease (cerebrovascular disease, previous stroke or peripheral vascular disease). Definitively treated cardiac disease, for example, previous myocardial infarction treated with stents or coronary artery bypass grafting with no ongoing symptoms is permitted.
► Abnormal thyroid function (hyperthyroidism or hypothyroidism).
► History of seizures, excluding febrile convulsions while an infant.
► Abnormal ECG with evidence of arrhythmia (except first degree heart block which has been stable for 3 months).
► Concomitant therapy with an excluded medication (see Concomitant therapies section).
► Glaucoma or raised intraocular pressure for any reason.
► Patients with established liver disease defined as Child-Pugh class B or C.
► Documented medical history of psychiatric disorders (excluding depression).
► History of drug dependence or addiction at any time.
► Female participant who is pregnant, lactating or planning pregnancy during the course of the trial.
► Receiving an investigational drug or biological agent within 6 weeks (or five times the half-life if this is longer) prior to study entry.

---

Assessment of Chronic Illness Therapy–Fatigue (FACIT-F) Questionnaire), quality of life scores (King's Sarcoidosis Questionnaire (KSQ), EuroQoL-5D-5L (EQ5D) and Short Form-36 (SF-36) questionnaires), depression symptom screening (Hospital Anxiety and Depression Score (HADS) questionnaire), sleep quality (Pittsburgh Sleep Quality Index (PSQI) questionnaire) and a custom costs questionnaire for health economic analysis. In addition, participants perform a modified shuttle walk test (MSWT) to determine exercise capacity and perform spirometry to measure baseline lung function. After these assessments have been completed, participants receive their first 2-week supply of study medication, either methylphenidate 10 mg two times per day or matched placebo.

The visit schedule over the first 6 weeks focuses on safety and adverse events, plus establishing a stable dose. After 1 week of therapy, participants receive a phone call from the study team to review potential adverse effects from the medication. A further week later, the participant is seen for their first follow-up visit with safety measures taken (full blood count, renal function and liver function plus ECG) and review of side effects. If no adverse events

have been identified then participants are titrated up to the higher dose of 20 mg methylphenidate two times per day if there has been insufficient clinical improvement in fatigue. Further safety visits (via phone on weeks 3 and 5, and with the study team for blood tests and ECG at week 4) subsequently occur; in the event of side effects emerging participants may be down-titrated to 10 mg methylphenidate two times per day (see Discontinuation criteria section for more details).

At 6 weeks, participants return for a further safety visit, undergoing the same blood tests and a repeat ECG. In addition, the study questionnaires are repeated (FAS, FACIT-F, KSQ, EQ5D, SF-36 and HADS), with the exception of the costs questionnaire which is performed at 12 and 24 weeks. Participants continue to receive phone calls to monitor for side effects every 2 weeks but are next seen face to face at 12 weeks, where all measures (safety bloods and ECG, adverse events, questionnaires including costs and PSQI, MSWT and spirometry) are repeated. In addition, participants wear a wrist-worn accelerometer for a further 7 days to review daily activity levels and answers questions on sleep quality.

A further visit is undertaken at 18 weeks where participants undergo the same investigations as at 6 weeks, and finally again at 24 weeks where all study assessments are repeated, including an additional exit questionnaire collecting information about experiences of the trial and the medication. Participants on the higher dose of methylphenidate (20 mg two times per day) are reduced to 10 mg two times per day for 2 weeks; participants who complete the study on the lower dose discontinue all study medications at 24 weeks.

Following discontinuation of the study medication participants will undertake all questionnaires, excluding the cost questionnaire, and be assessed for side effects 6 weeks after discontinuing medications (30 weeks after commencing study medications). This will allow investigation of whether fatigue returns to baseline level following discontinuation of medication, or whether any benefit compared with baseline persists.

All participants are asked at the point of consenting to enter the study whether they would be willing to be approached about participating in focus groups following their completion of the study. Those who agree are offered the option to participate in focus groups following the completion of their trial participation; this is to investigate their experience and perception of the trial. There is a standardised topic guide to direct discussions (available from the authors). All data are audio recorded and then transcribed. It is not a mandatory part of the study; we aim to have three groups of six to eight participants at completion of the study.

## Outcomes

Primary and secondary outcomes are listed in box 2. Data collection points for each of the outcome measures are shown in table 1. The primary outcomes are related to feasibility, based on recruitment (including number of

---

**Box 2  Primary and secondary outcomes**

**Primary outcome: Feasibility assessments**
► Recruitment rate and retention of participants.
► Reason for exclusion or withdrawal from the study.
► Number of participants suffering side effects or requiring reduction of methylphenidate dose due to emergence of side effects and the dose tolerated by greatest proportion of participants.
► Indication(s) of continuation of effect at stable dose during the treatment period.
► Number of patients correctly using accelerometers.
► Acceptability of number of study visits and assessments.
► Acceptability of randomisation.
► Acceptability of receiving a controlled drug.
► Mean change in fatigue score and SD of change (for subsequent power calculations).
► Patient perception of trial involvement (focus groups and exit questionnaire).

**Secondary outcomes: Exploratory assessment of clinical effects of methylphenidate over 24 weeks**
► Change in fatigue scores—Fatigue Assessment Scale and Functional Assessment of Chronic Illness Therapy-Fatigue.
► Disease-specific quality of life—Kings Sarcoidosis Questionnaire.
► Depression and anxiety—Hospital Anxiety and Depression Scale.
► Generic quality of life—EuroQoL-5D-5L and Short Form-36.
► Usage of health and social care — cost questionnaire.
► Sleep quality—Pittsburgh Sleep Quality Index; augmented with results from wrist-worn accelerometer data (awakenings).
► Modified shuttle walk test distance—number of shuttles completed.
► Physical activity in free-living—mean accelerometer output (most active 5 hours, time in moderate or vigorous activity and time in sedentary behaviours).
► Lung function—spirometry (forced expiratory volume in 1 s and forced vital capacity).

---

patients who are eligible to participate), the number of participants that are retained at each point of the study, the number of participants providing data through to the end of the trial and an estimate of treatment effect using change in the fatigue score measurements (FAS and FACIT-F) between methylphenidate and placebo groups which will influence future power calculations.

The output from this trial will influence the choice of primary outcome for future trials investigating sarcoidosis-associated fatigue. This is likely to be the FAS questionnaire as it is both symptom-specific for fatigue and has been validated in sarcoidosis cohorts previously.[20] However, the interaction between changes in fatigue scores (both FAS and FACIT-F) and other measures, including depression and anxiety (HADS) and quality of life (KSQ, EQ5D and SF-36), will be reviewed to determine how these measures change longitudinally with change in fatigue.

Additionally, a cost questionnaire will be administered to all participants. This is a custom-made instrument being piloted in this study which will look at costs incurred on a day-to-day basis by participants, including changes in out-of-pocket costs to participants relating to their health, health service usage and employment/

working time. These methods can then be amended as required for appropriate cost estimation in a definitive trial in the future.

## Concomitant therapies

Participants are able to receive any immunosuppressant drugs that would treat their sarcoidosis, but they are required to be on a stable dose of their medication for 6 weeks prior to study enrolment with no planned change in dose during the trial. In the event that treatment is altered, either reducing or increasing the dose of existing therapy or starting additional medication for their sarcoidosis, this is recorded in their case report file.

There are a number of drugs which are contraindicated when receiving methylphenidate due to the risk of side effects; participants are not permitted to receive these during the trial. These include any tricyclic antidepressant (TCAs); any monoamine oxidase inhibitor (MAOI); all antipsychotics; sympathomimetics including phenylephrine, ephedrine or pseudoephedrine; buprenorphine; tramadol; levodopa; clonidine; methylene blue; warfarin. Although both TCAs and MAOIs are prohibited, selective serotonin inhibitors are allowed and have been used alongside methylphenidate safely previously.[21]

## Safety considerations

Methylphenidate is associated with side effects due to cardiac, neurological or psychiatric effects. The trial has been designed to minimise the risk of these complications occurring while receiving methylphenidate, both through the exclusion criteria and the visit schedule with safety monitoring. Cardiac side effects are screened for at baseline with ECG monitoring, and at every study visit thereafter. Any development of cardiac complications or ECG abnormalities, even if clinically asymptomatic, results in discontinuation of the study medication. Neurological and psychiatric problems are screened for, with any history of seizures (aside from febrile convulsions during childhood) or psychiatric disease (except depression) preventing patients from participating in the trial. Participants who are included in the trial are monitored at each visit for the emergence of any side effects, including seizures, abnormal behaviour or personality change.

A final safety consideration relates to methylphenidate being a controlled drug. The potential illicit use of methylphenidate risks participants selling or giving away medications to others. All participants will be informed about the illegal nature of these activities, and asked to sign a document to indicate their understanding and agreement. Medication prescriptions are initially for 2 weeks, increased to 6 weeks after the third visit (at 6 weeks) with participants bringing their remaining supply of medication to each visit for pill-count checking.

## Discontinuation criteria

Immediate discontinuation will occur, regardless of the present dose, if participants develop a generalised rash or pruritus considered to be related to the intervention, develops an abnormality on ECG, suffers chest pain consistent with angina, experiences severe anxiety or euphoria, personality change or bizarre behaviour or if any psychiatric disease manifests during the trial. Other reasons for immediate discontinuation are seizures and severe hypertension (defined as a systolic blood pressure >180 mm Hg on two separate occasions or presenting with any features of malignant hypertension). Side effects which will lead to down-titration of the study drug in the first instance (if the participant is receiving the higher dose of 20 mg methylphenidate two times per day) include nervousness or restlessness, significant nausea or indigestion, nasal stuffiness, cough, arthralgia or anorexia. Participants can also choose to discontinue the study medication at any time for any reason. In consenting to the trial participants are agreeing to trial treatments, trial follow-up and data collection, including storage of anonymised data within a dataset available to other researchers from a data repository at the end of the trial.

## Sample size and recruitment

The sample size of 30 patients was chosen in order to answer questions regarding recruitment, number of eligible participants and reasons for exclusion and to pilot some elements of the trial, including the use of accelerometers measuring activity levels as part of the outcomes. This also ensures a minimum of 12 participants per group as per recommendations.[22] This study is not powered to identify clinical efficacy, although an estimation of effect size between methylphenidate and placebo will enable power calculations to be performed for any future follow-on studies investigating methylphenidate.

All research activities occur at a single tertiary hospital site, although within this study we are using PICs to improve recruitment by enabling a wider number of patients to be considered for inclusion within this study.

## Data management and monitoring

A purpose-designed, secure password-protected electronic case report form is used for data entry throughout the trial, complying with data protection requirements on confidentiality and anonymity. The database enables screened patients to be randomised, linking with the local hospital pharmacy to securely randomise patients via email. This maintains blinding of the clinical team, research team and investigators and patients. Allocation is determined based on block randomisation in blocks of five with stratification by severity of fatigue (FAS 22–34 and FAS 35–50). The system is managed by the Norwich Clinical Trials Unit and can only be accessed by the chief investigator or appropriate members of the study team with delegated responsibility.

Throughout the trial monitoring will occur quarterly, carried out locally with oversight from the clinical trials unit and sponsor. A trial steering committee and safety committee monitor trial performance and adverse events. The two groups meet 6 monthly and may terminate the

trial if significant concerns become apparent, although formal stop criteria have not been set.

## Data analysis

Initial data analysis will evaluate prespecified feasibility criteria. First, a recruitment rate of 10% from our local sarcoidosis cohort is required to meet our recruitment aims; in the event of lower recruitment rates then consideration will need to be given to the ability to recruit the necessary number of sites, although this will be determined by the estimate of the mean difference in fatigue scores (effect size). The mean difference will be estimated using a general linear model for continuous outcomes, with an emphasis on estimation rather than hypothesis testing. Analysis will be based on an intention-to-treat approach. Adverse event data will be tabulated by category per group. No formal comparison will be made. Previous trails of methylphenidate have suggested an adverse events rate of less than 5% but these have not used the drug for more than 12 weeks (8 weeks in the previous trial involving patients with sarcoidosis-associated fatigue[17]). An adverse event rate of more than 10% during this 24-week trial would trigger consideration of the safety of methylphenidate for longer periods and the suitability of using this agent. Finally, previous trials have managed a retention rate of more than 80% of participants; we will review the retention rate over this longer trial, aiming for over 60% retention to suggest a larger trial over such a period would be feasible. Understanding whether the other clinical measures being employed here (including the MSWT and accelerometer measure of activity) are responsive to changes in fatigue scores, and therefore worth including in subsequent trials, will determine if they should be included in future studies.

Power calculations for any subsequent trial will be performed using the data from change in the FAS score, which will inform sample size requirements, as well as data on drop-out rates through the trial.

Participant perception of undertaking the trial will be reviewed using the discussions in the focus groups. Participant views on the number of study visits, the intervention received (including the fact that it is a controlled drug), the specific measures used including the MSWT and activity monitor, and overall impression of participating in the trial (both positive and negative) will be reviewed following the focus groups, which will act as a debrief session for participants. This qualitative process evaluation has been recommended for use in the development of some RCTs and will enable optimisation of a future RCT through the experiences of participants.[23]

Exploratory analysis of the data will take place to investigate whether any change in fatigue or other measures is seen during the trial between groups, with appropriate adjustment for baseline values. The study is not powered for these clinical outcomes; this will be clearly stated in any outputs from these analyses.

## Study registration and approvals

Approval from the Medicines and Healthcare products Regulatory Agency (MHRA - EudraCT reference number 2016-000342-60) and Research Ethics Committee (REC - East of England; Cambridge Central, reference number 16/EE/0087) has been gained, with the trial adopted by the Health Research Authority (HRA). The study has been registered on a clinical trial database (http://clinicaltrials.gov), reference number NCT02643732 (pre-results). The study has been open for recruitment since October 2016 and will continue until July 2018.

## DISCUSSION

The FaST-MP study has been developed to inform decision-making regarding any future trial investigating the use of neurostimulants for the treatment of sarcoidosis-associated fatigue, a frequent and often challenging manifestation of chronic sarcoidosis. The results from this study will allow us to determine if progressing to a full-size trial using this drug is feasible, how many centres would be required, how the study would need to be structured and which outcome measures would be best employed to monitor progress throughout the trial. Piloting new outcome measures (costs questionnaire and activity monitors) during the course of this RCT will also enable us to evaluate these measures and their suitability for future studies.

**Contributors** CA, RF, AS, ABC, APJ and AMW: all made substantial contributions to the conception and design of the study. CA: wrote the manuscript drafts. AMW: made significant revisions to the manuscript. All authors read, amended and approved the final manuscript.

**Funding** This work is an independent research supported by the National Institute of Health Research (NIHR Doctoral Research Fellowship, Dr Chris Atkins, DRF-2015-08-190).

**Disclaimer** The views expressed in this publication are those of the authors and not necessarily those of the NHS, the National Institute for Health Research or the Department of Health.

**Competing interests** None declared.

**Patient consent** Obtained.

**Provenance and peer review** Not commissioned; externally peer reviewed.

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
