## [Reviewer comments · BMJ Open]

ARTICLE DETAILS

TITLE (PROVISIONAL)	Feasibility study of a randomised controlled trial to investigate the treatment of sarcoidosis-associated fatigue with methylphenidate (FaST-MP): a study protocol
AUTHORS	Atkins, Christopher; Fordham, Richard; Clark, Allan; Stockl, Andrea; Jones, Andrew; Wilson, Andrew

VERSION 1 – REVIEW

REVIEWER	Professor dr Violeta Vucinic University Clinic of lung diseases, Clinical Center of Serbia, Belgrade, Serbia Medical School, University of Belgrade, Serbia
REVIEW RETURNED	26-Jul-2017

GENERAL COMMENTS	Congratulations. Good work. I am eagerly waiting to read the final study results.
---

REVIEWER	Manuel L. Ribeiro Neto Cleveland Clinic, USA
REVIEW RETURNED	04-Oct-2017

GENERAL COMMENTS	I was pleased to see that the authors are conducting this study entitled "Feasibility study of a randomised controlled trial to investigate the treatment of sarcoidosis-associated fatigue with methylphenidate (FaST-MP): a study protocol. This topic is extremely relevant to sarcoidologists and patients with sarcoidosis. This will be a feasibility study of a randomized controlled trial, so it is an important first step. The 3:2 allocation is appropriate and they are preserving allocation concealment and blinding to the best of their abilities. The outcomes are very appropriate, objective, using well validated questionnaires. The follow up time is appropriate. Regarding the inclusion criteria, I agree with including patients with "treatment unchanged for 6 weeks". This will probably exclude some patients in the very acute and inflammatory phase of the disease, where the fatigue can improve with immunosuppression or time alone (SARCOIDOSIS VASCULITIS AND DIFFUSE LUNG DISEASES 2016; 33: 124-129). However, I have two criticisms. First, the inclusion criteria "without anticipation of change in the treatment during the trial period" is very subjective and difficult to predict. Not sure if it will be beneficial to keep it.
---

	And second, it may be better to define a time period from the diagnosis to the inclusion in the study (e.g. 3 months, 6 months), to exclude patients that would improve spontaneously or with immunosuppression alone. Minor comment: Page 7, line 50: there is a misplaced parenthesis after the number 21 Overall, I applaud the authors for this work. It will bring important information to the Sarcoidosis literature. And it will hopefully improve our ability to help patients with Sarcoidosis-related fatigue.
--	---

REVIEWER	Anoop M. Nambiar, MD, MS University of Texas Health Science Center San Antonio, South Texas Veterans Health Care System, USA
REVIEW RETURNED	07-Oct-2017

GENERAL COMMENTS	Atkins and investigators propose a pilot placebo-controlled RCT to support the design of a future, larger, prospective trial evaluating the safety and efficacy of methylphenidate for sarcoidosis-associated fatigue. Overall, this is a well-written protocol with appropriate study design, inclusion/exclusion criteria, primary/secondary outcomes, and safety measures/endpoints. However, I have a few questions and/or concerns that I'm hoping the authors can address:  1. Since fatigue may affect those with extrapulmonary more than pulmonary disease (Ref: Fleischer et al., Resp Care 2014), I would recommend that each patient's sarcoid manifestations be carefully documented. 2. The inclusion criteria of "stable" disease as defined as unchanged treatment for 6 weeks appears subjective and arbitrary. Significant fatigue can still be present in those with subclinical disease. I think it would be of interest to include biomarkers such as circulating TNF-alpha and CRP (Drent M, et al. ERJ 1999) and PET scans to evaluate their potential utility as an exploratory outcome measure in a larger prospective study. 3. Careful documentation of specific treatments for sarcoidosis is essential as prednisone use has been associated with increased fatigue in these patients. 4. Why the complex 3:2 randomization scheme? If unequal arm size is desired, why not 2:1 (20 on study drug, 10 placebo) to simplify things a bit? 5. If this trial was approved by the local ethics board in June 2016 and is still presently open for enrollment, how many patients have already been recruited? It is important to clarify why recruitment for this study has been slow since fatigue is felt to be present in nearly 80% of sarcoidosis patients. 6. I would like to know the authors' opinion on why there has been little progress made using methylphenidate (or similar) for sarcoidosis-associated fatigue when a pilot RCT with a favorable safety profile and promising results was published in 2008, or nearly a decade ago?
--

	7. Although methylphenidate has not been studied prospectively in sarcoidosis-associated fatigue for longer than 8 weeks, is there any actual published data that prolonged use of methylphenidate can cause significant harm? In the setting of a past positive pilot study, is there a real need for another pilot study? Why not propose this study to be a multicenter, phase 2 RCT? If positive, then this would be steps closer to being an approved treatment.
--	--

VERSION 1 – AUTHOR RESPONSE

Reviewer 1 comments (Professor Violeta Vucinic)

Many thanks for the positive comments regarding our study. We hope to have the final study results at the end of next year.

Reviewer 2 comments (Manuel L. Ribiero Neto)

Thank you for your positive comments on the study design. We note the comments made about the difficulty of anticipating change in treatment for sarcoidosis during the trial and agree with this; this wording has been used so that we are not including patients where they are not on stable treatment for their sarcoidosis. We do accept that it may not be beneficial to keep this statement, but we were keen to exclude patients where fatigue stemmed from high dose steroid use which would be decreasing during the study. As this is part of the current inclusion and exclusion criteria we have kept this for now, but it will be reviewed when considering future studies.

The second comment, regarding a minimum duration of time since the diagnosis of sarcoidosis, was also considered. The reason that this has not been included is to allow inclusion of patients where diagnosis has been delayed but the disease has likely been present for a much longer period of time (for example, due to delay in seeking medical help or slow referral to secondary care for the diagnosis). We accept that specifying a minimum duration would help to minimise the risk of including patients where the fatigue will spontaneously improve and will review this for future studies.

Reviewer 3 comments (Dr Anoop M Nambiar MD MS)

We thank Dr Nambiar for his positive comments on the protocol, and for his questions regarding the study. I have tried to address the specific questions below –

1. Since fatigue may affect those with extrapulmonary more than pulmonary disease, I would recommend that each patient's manifestations be carefully documented.

Response: This is an important point and one that we have now clarified within the text (page 7) – we have been collecting this data from all patients being screened for the study, and for all participants entering the study.

2. The inclusion criteria of “stable” disease as defined as unchanged treatment for 6 weeks appears subjective and arbitrary – it would be of interest to include biomarkers and PET scans to evaluate their potential utility as an exploratory outcome measure in a larger prospective study.

Response: We agree that the criteria is subjective, but has been chosen as a pragmatic marker of patients for whom fatigue is an issue but no other evidence of active disease that would necessitate the use of immunosuppression is present, hopefully emulating the clinical situation that clinicians would find themselves in when considering the use of neurostimulants. We do not believe biomarkers would provide additional information within this feasibility study as there would not be enough participants to determine if these biomarkers could have a role in a larger study.

We agree that the inclusion of these biomarkers may be of interest in a much larger study where differences in these biomarkers between groups could be determined.

3. Careful documentation of specific treatments for sarcoidosis is essential as prednisone use has been associated with increased fatigue.

Response: We completely agree with this point and have been collecting this data on all patients screened for the study, including all those who participate in the study. We collect data on all drugs and doses given for sarcoidosis (steroids and other immunosuppressants) for everyone participating and have now clarified this on page 7 of the manuscript.

4. Why the complex 3:2 randomisation scheme? If unequal arm size is desired, why not 2:1 to simplify things a bit?

Response: A 3:2 randomisation scheme was chosen to increase the number of participants receiving methylphenidate, so that we would have more data on drug efficacy and tolerability/safety over the trial duration. A 2:1 randomisation scheme was considered but previous work had suggested that it was preferable to aim for at least 12 patients in either arm; a 3:2 randomisation in a sample of 30 patients fulfilled this brief. After discussion with Dr Clark, the study statistician, it was also felt to be appropriate using a block-randomisation method with blocks of 5 (3 treatment, 2 placebo).

5. If this trial was approved by the local ethics board in June 2016 and is still presently open for enrollment, how many patients have already been recruited? It is important to clarify why recruitment for this study has been slow since fatigue is felt to be present in nearly 80% of sarcoidosis patients. Ethical approval was granted in June 2016 but delays in securing IMP and opening the study meant that the study opened in October (page 15 in the protocol manuscript) and has been recruiting since. Recruitment has been less than expected, however this data will be presented as part of the feasibility data when the study results are published rather than being discussed at this point.

6. I would like to know the authors' opinion on why there has been little progress made using methylphenidate (or similar) for sarcoidosis-associated fatigue when a pilot RCT with a favourable safety profile and promising results was published nearly a decade ago?

Response: We are unclear why further data has not been forthcoming in the decade since Dr Lower and colleagues published their cross-over trial of methylphenidate for treatment of fatigue. We have used the medication in some sarcoidosis patients with debilitating fatigue and have waited for further evidence to be released in the literature. It was the lack of follow-up work to the small that precipitated our decision to undertake this study.

7. Although methylphenidate has not been studied prospectively in sarcoidosis-associated fatigue for longer than 8 weeks, is there any actual published data that prolonged use of methylphenidate can cause significant harm? In the setting of a past positive pilot study, is there a real need for another pilot study?

Response: Whilst methylphenidate has been used for many years with a good safety profile, there is a real lack of longer-term data in patients with sarcoidosis. We were unsure if an 8-week period would give a realistic reflection of its effect and tolerability in patients with sarcoidosis. We were also unclear how many patients with sarcoidosis-associated fatigue would be eligible for treatment – a large number of patients with sarcoidosis report fatigue, but no data exists for the number who would be suitable for treatment.

We hope that these changes are satisfactory to the editorial team and reviewers, and will allow the protocol to proceed to publication.

VERSION 2 – REVIEW

REVIEWER	Anoop Nambiar UT Health San Antonio
REVIEW RETURNED	08-Nov-2017
GENERAL COMMENTS	No additional comments since my initial review. Authors satisfactorily addressed my concerns.